# Catastrophic chromosomal restructuring during genome elimination in plants

Ek Han Tan[1,2], Isabelle M Henry[1,2], Maruthachalam Ravi[3], Keith R Bradnam[2], Terezie Mandakova[4], Mohan PA Marimuthu[1,2], Ian Korf[2,5], Martin A Lysak[4], Luca Comai[1,2]*, Simon WL Chan[1,6]†

[1]Department of Plant Biology, University of California, Davis, Davis, United States; [2]Genome Center, University of California, Davis, Davis, United States; [3]School of Biology, Indian Institute of Science Education and Research, Thiruvananthapuram, India; [4]Central European Institute of Technology, Masaryk University, Brno, Czech Republic; [5]Department of Molecular and Cellular Biology, University of California, Davis, Davis, United States; [6]Gordon and Betty Moore Foundation, Howard Hughes Medical Institute, University of California, Davis, Davis, United States

**Abstract** Genome instability is associated with mitotic errors and cancer. This phenomenon can lead to deleterious rearrangements, but also genetic novelty, and many questions regarding its genesis, fate and evolutionary role remain unanswered. Here, we describe extreme chromosomal restructuring during genome elimination, a process resulting from hybridization of *Arabidopsis* plants expressing different centromere histones H3. Shattered chromosomes are formed from the genome of the haploid inducer, consistent with genomic catastrophes affecting a single, laggard chromosome compartmentalized within a micronucleus. Analysis of breakpoint junctions implicates breaks followed by repair through non-homologous end joining (NHEJ) or stalled fork repair. Furthermore, mutation of required NHEJ factor DNA Ligase 4 results in enhanced haploid recovery. Lastly, heritability and stability of a rearranged chromosome suggest a potential for enduring genomic novelty. These findings provide a tractable, natural system towards investigating the causes and mechanisms of complex genomic rearrangements similar to those associated with several human disorders.

*For correspondence: lcomai@ ucdavis.edu

†Deceased

**Reviewing editor**: Bernard de Massy, Institute of Human Genetics, CNRS UPR 1142, France

## Introduction

Nucleosomes containing variant histone (centromeric histone H3, CENH3) (*Verdaasdonk and Bloom, 2011*) (also known as CENP-A) determine centromeres. In the absence of the endogenous CENH3, *Arabidopsis thaliana* mitotic and meiotic functions can be complemented by chimeric CENH3 (*Ravi and Chan, 2010*; *Ravi et al., 2010*) or CENH3 from diverged plant species (*Maheshwari et al., 2015*), but crossing these strains to wild-type individuals results in frequent loss of the chromosomes marked by the variant CENH3. Following stochastic genome elimination in the early mitotic divisions, the progeny can be haploid, aneuploid or diploid (*Ravi and Chan, 2010*; *Ravi et al., 2014*). In nature, similar phenomena involve defective CENH3 loading (*Sanei et al., 2011*). Thus, mating of individuals that express diverged CENH3s, can lead to mitotic catastrophe.

The consequences of mitotic malfunction on genome integrity can be dire (*McClintock, 1984*; *Gordon et al., 2012*). Missegregated chromosomes can lead to aneuploidy (*Janssen et al., 2011*), but also to extensive and catastrophic restructuring resulting in, sequentially, chromosome sequestration in micronuclei, endonucleolytic damage, defective repair, and finally rescue (*Crasta et al., 2012*; *Hatch et al., 2013*; *Zhang et al., 2013*). The resulting structurally rearranged chromosomes may contribute to cancer or developmental syndromes (*Hastings et al., 2009*;

**eLife digest** The genome of an individual organism contains all the instructions needed to build and maintain that individual. Any changes to the DNA in the genome can alter the instructions that are given to cells, which can lead to cancer and other diseases. However, changes to the genome can sometimes be beneficial as they can introduce more variety into the instructions carried by different individuals, which increases their potential to adapt to changes in their environment.

In plants and animals, DNA is arranged into structures called chromosomes. Generally, an individual's genome contains two copies of each chromosome; one inherited from their mother and one from their father. However, occasionally during reproduction, all the chromosomes from one of the parents are left out from the cells of the offspring in a process called 'genome elimination'. This makes individuals that carry only half the normal number of chromosomes, known as haploids. Sometimes the process of genome elimination is disrupted, which leads to individuals that have incomplete genomes or chromosomes that carry big rearrangements of the DNA, as if they had been shattered and put back together incorrectly.

In a small plant known as *Arabidopsis thaliana*, genome elimination frequently happens in the offspring of two individuals that carry different versions of a gene called centromeric histone H3 (CENH3). However, it is not clear how this works, or what roles genome elimination plays in evolution and disease.

Here, Tan et al. studied genome elimination by cross-breeding *Arabidopsis* plants that carried a mutant form of CENH3 with plants that have a normal version of the protein. The experiments found that many of the offspring were haploid. Some of the others carried an extra copy of an entire chromosome or a section of a chromosome. A third group had an extra copy of a chromosome that was missing some sections or had been rearranged. These 'shattered' chromosomes were always formed from chromosomes that came from the parent plant with a mutant form of CENH3.

Tan et al. also found that a protein called DNA Ligase 4, which helps reconnect broken DNA strands, is involved in repairing the breaks in these shattered chromosomes. Some of the genetic rearrangements documented in the experiments were passed on to subsequent generations of plants, which suggests that these genomic changes can be stable enough to be inherited.

The genomic rearrangements observed in the *Arabidopsis* plants are similar to those seen in patients with cancer and other genetic diseases. Tan et al. findings show that *Arabidopsis* plants provide a useful system for studying these genome rearrangements, which may inform efforts to treat these human diseases.

*Liu et al., 2011*; *Stephens et al., 2011*; *Jones and Jallepalli, 2012*). Nevertheless, chromosomal rearrangements are not necessarily deleterious: some may influence fitness by altering recombination or gene dosage (*Comai et al., 2003*). It is possible that pathways leading to disease and to diversity share a common mechanistic basis (*Zhang et al., 2013*). Genome elimination in *Arabidopsis* provides a previously lacking organismal system to investigate genome instability during mitotic catastrophes, connected mechanisms, and consequences.

## Results

We used the *GFP*-tailswap haploid inducer (*Ravi and Chan, 2010*; *Ravi et al., 2014*) in the experimental setup illustrated in *Figure 1*. This strain is in the Col-0 background and carries a homozygous *CENH3* null mutation whose function is partially complemented by a chimeric CENH3 in which an N-terminal *GFP* fused to the H3.3-like N-terminal tail replaces the native CENH3 N-terminal tail. We crossed this strain to polymorphic accession L*er gl1-1* to track haplotypes in the F1 progeny and obtained the expected haploid induction frequency (*Ravi and Chan, 2010*; *Ravi et al., 2014*) (*Figure 1*). The recessive *gl1-1* mutation confers trichomeless leaves in paternal L*er gl1-1* haploids while it is masked in Col/L*er* diploid hybrids. We sequenced 10 of the phenotypically diploid Col/L*er* individuals with wild-type phenotype, performed dosage plot and single nucleotide polymorphism (SNP) analysis and found that 100% of these were diploid with 50% Col and L*er* genomes respectively (*Figure 1—figure supplement 1*). Plants from the aneuploid class exhibited multiple pleiotropic and morphological defects and had trichomes, except in the rare exception when

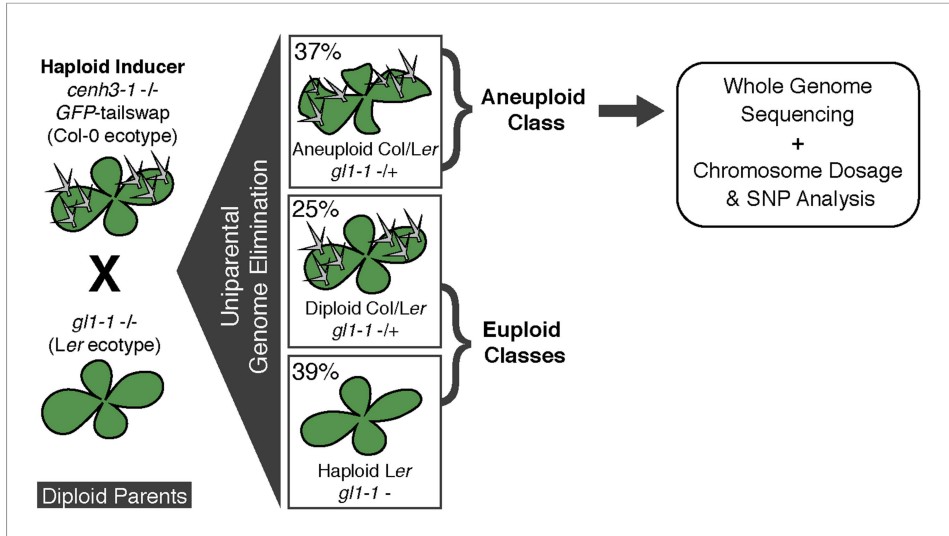

**Figure 1**. Ploidy types resulting from centromere-mediated uniparental genome elimination. The altered CENH3 'GFP-tailswap' strain was hybridized to the recessive glabrous1-1 mutant. Mean percentages of haploid, diploid and aneuploid progeny obtained from crosses to three independent GFP-tailswap lines are indicated, as determined after phenotypic characterization. Individuals belonging to the aneuploid class were sequenced and subjected to chromosome dosage and single nucleotide polymorphism (SNP) analysis as indicated by the arrow.
The following figure supplement is available for figure 1:

**Figure supplement 1**. Dosage plots and SNP analysis of diploids from GFP-tailswap haploid induction crosses.

the GL1 locus was lost. The five recognizable primary trisomic (2n + 1) phenotypes were represented (**Steinitz-Sears, 1963**; **Koornneef and Vanderveen, 1983**): Chromosome 1 (Chr1) trisomics have dark green, serrated leaves and are dwarfed, Chr2 trisomics exhibit round leaves and are late flowering, Chr3 trisomics have narrow, yellow green leaves, Chr4 trisomics display narrow and smaller flat leaves, and Chr5 trisomics display light green and narrow leaves. However, aneuploid plants with more severe or unusual phenotypes were also observed, suggestive of other chromosomal combinations or more serious chromosomal aberrations. Chromosome dosage analysis based on whole genome sequencing (**Henry et al., 2010**) (**Supplementary file 1**) distinguished three chromosomal alteration types in aneuploids (**Figure 2**). Similar outcomes were obtained using independently derived haploid inducers, either expressing GFP-tailswap (**Figure 2E** and **Figure 2—figure supplement 1**) or CENH3 from other plant species (**Maheshwari et al., 2015**). The most common type, numerical aneuploids, display whole chromosome aneuploidy such as in the classical primary trisomics (**Figure 2B** shows an example for a numerical Chr3). In our dataset, single primary trisomics (2n + 1) account for 75% of the numerical class. Other individuals from the numerical class with two or more extra whole chromosomes included 16% double primary trisomics (2n + 1 + 1), 2% triple primary trisomics (2n + 1 + 1 + 1) and 3% quadruple primary trisomics (2n + 1 + 1 + 1 + 1). Additionally, we obtained disomic Chr4 haploids (n + 1, a type of numerical aneuploidy that were not included in this analysis) as well as Chr2 or Chr3 monosomic diploids (2n − 1) at 4% frequency (**Figure 2—figure supplement 2**). These have never been described in Arabidopsis before, possibly because, if they were to arise from meiotic defects, they would result from nullisomic gametes, which are not viable (**Henry et al., 2009**). Aneuploids resulting from mitotic failure do not have those constraints.

The second alteration type is defined by simple truncations and repair of at most two double stranded DNA breaks per chromosome (**Figure 2C** shows an example of truncated Chr3). This truncated class was found to occur in 22% of the aneuploid population. In the third class, a single chromosome exhibited many oscillations in copy number state, as if shattered and subsequently rearranged (**Figure 2D** shows an example of shattered Chr3). This shattered class was found to occur in 11% of the aneuploid population. Additionally, some of the aneuploids exhibited a combination of

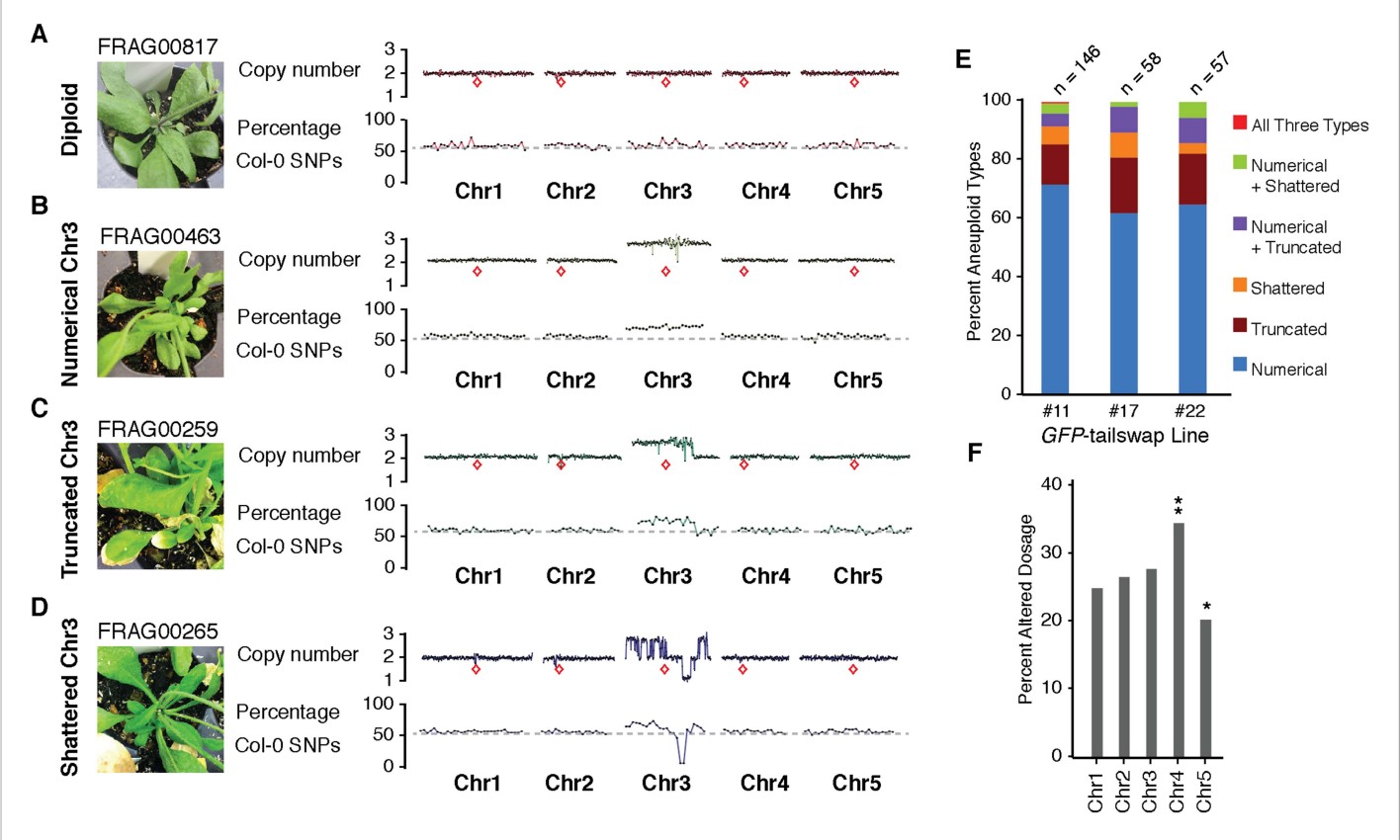

**Figure 2**. Characterization of the three distinct aneuploid types from *GFP*-tailswap haploid induction crosses. (**A–D**) Dosage plots from all five *Arabidopsis* chromosomes in consecutive non-overlapping 100 kbp bins and the corresponding SNP plot for the % haploid inducer genome (Col-0) present in each sample. A diploid Col/L*er* hybrid (**A**), an individual with primary Chr3 trisomy from the numerical aneuploid class (**B**), an individual with a truncated trisomic Chr3 (**C**) and an individual with shattered Chr3 (**D**) are shown here. Centromere positions are indicated by red diamonds. (**E**) Percentages of the different aneuploid types obtained from three different *GFP*-tailswap haploid inducer lines. (**F**) For each chromosome, the percentage of aneuploid individuals exhibiting altered dosage for that particular chromosome is plotted. All aneuploids characterized in this study are included. Chr4 is overrepresented (**Student's *t*-test, p < 0.01) while Chr5 is underrepresented (* Student's *t*-test, p < 0.05).

The following figure supplements are available for figure 2:

**Figure supplement 1**. Copy number and T-DNA positions of the *GFP*-tailswap transgene in three independently derived transgenic lines.

**Figure supplement 2**. Dosage plots and SNP analysis of atypical aneuploids from *GFP*-tailswap haploid induction crosses.

**Figure supplement 3**. Diversity of primary trisomic aneuploids derived from a selfed triploid (Col-0 ecotype).

**Figure supplement 4**. Representative dosage plots from 96 individuals from a selfed *GFP*-tailswap haploid inducer.

numerical, truncated and shattered chromosome types (*Figure 2E*). Alteration of copy number for Chr1, 2 and 3 are represented at similar frequencies based on the average copy number alteration of all five chromosomes, with Chr4 and 5 alterations being, respectively, over- and under-represented (*Figure 2F*). This may be explained by the uneven distribution between chromosomes of few, selected genes that are highly dosage-sensitive. According to this hypothesis, Chr4 would be selectively depleted for such genes.

Chromosomal truncations have been reported from a selfed trisomic (*Huettel et al., 2008*). To assess whether truncated and shattered aneuploid types could be produced from meiotic missegregation, we sequenced 96 individuals produced by a selfed Col-0 triploid. Because of the

irregular meiosis, most gametes produced by triploids are aneuploid (*Henry et al., 2007*). Dosage analysis revealed that all were numerical aneuploids (*Supplementary file 2* and *Figure 2—figure supplement 3*). To assess whether truncation and shattering could be the result of meiotic defects in the *GFP*-tailswap line, we sequenced 96 individuals from selfed *GFP*-tailswap and observed that 98% (n = 94) of the progeny were diploid while two individuals carried single primary trisomies of Chr2 and Chr3 respectively, representing only the numerical class of aneuploids (*Supplementary file 3* and *Figure 2—figure supplement 4*). Based on these results, we believe that truncated and shattered aneuploid classes from our crosses reflect genomic instability associated with mitotic errors in the early embryo.

Shattered chromosomes can be recovered from all five *A. thaliana* chromosomes (*Figure 3A*). In some cases, shattering appears to extend to two chromosomes (top panel of *Figure 3A*) only because the haploid inducer used carries a reciprocal Chr1/Chr4 translocation originating from the integration of *GFP*-tailswap T-DNAs. SNP analysis demonstrates that all duplicated (copy number 3) and triplicated (copy number 4) regions originated from the haploid inducer (*Figure 3B*). Single-copy regions displaying loss of heterozygosity carry L*er* alleles (i.e., wild-type), consistent with the loss of the haploid inducer haplotype.

Although aneuploids from the shattered class were often sterile, line FRAG00062 was partially fertile and allowed us to investigate the inheritance and stability of the variant DNA. We sequenced 16 F2 progeny from FRAG00062 and obtained two individuals with precisely the same shattered pattern as the F1 parent and 14 that appeared diploid (*Figure 4A*). Meiotic co-inheritance of all dosage variant segments is consistent with a single, stable chromosomal unit that was formed after a catastrophe. To confirm this hypothesis, we used DNA fluorescence in situ hybridization to visualize the FRAG00062 chromosomes using Col-0 derived BAC painting probes specific for Chr1 and Chr4 (*Figure 4B*). Mitotic cells contained 11 chromosomes (*Figure 4D*). FRAG00062 came from a cross using *GFP*-tailswap line #11, which carried a reciprocal Chr1/4 translocation (*Figure 2—figure supplement 1*). This allowed us to distinguish the haploid inducer Chr1, the L*er* Chr1, and a third Chr1 with rearranged signals, which we interpret as the shattered extranumerary chromosome (*Figure 4C* and *Figure 4—figure supplement 1*). During meiotic Metaphase I (*Figure 4D*) or other meiotic stages observed from male meiocytes (*Figure 4—figure supplement 2*), the shattered chromosome does not pair with the parental Chr1s.

Next, we sought to investigate why shattering is restricted to a single chromosome. During genome elimination crosses in other plant species, micronuclei are commonly observed

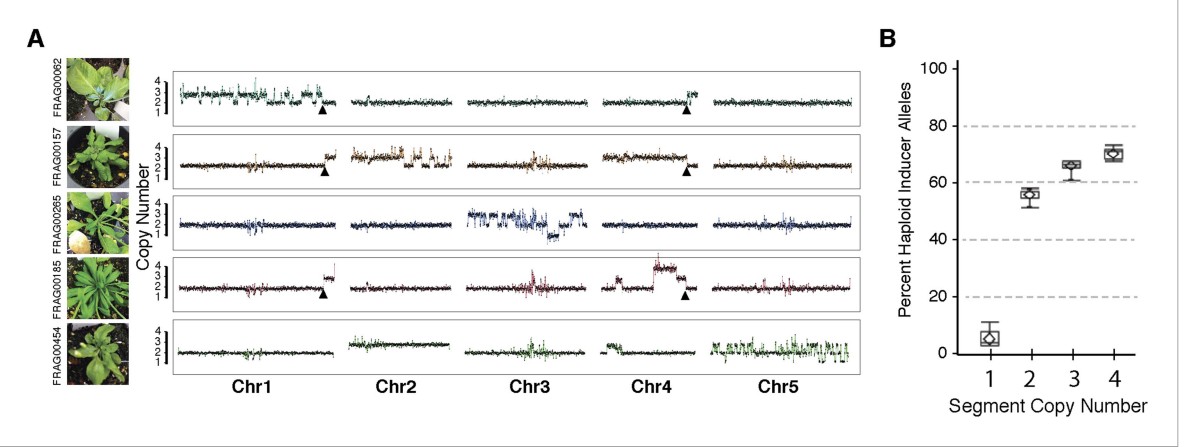

**Figure 3**. Shattered chromosomes are confined to a single chromosome originating from the haploid inducer. (**A**) Chromosome dosage plots based on non-overlapping 25 kbp bins across each chromosome for five aneuploid individuals with shattered chromosomes. The *GFP*-tailswap transgene insertion event that resulted in a reciprocal translocation between Chr1 and Chr4 in one of the haploid inducer parent (*GFP*-tailswap #11) is indicated with black arrowheads. The translocation is only visible in individuals for which chromosomes 1 and 4 are not balanced with each other. Duplications (copy number 3), triplications (copy number 4) as well as deletions accompanied with loss of heterozygosity (copy number 1) were observed from dosage plots. (**B**) Box plots of the percentage of haploid inducer genome present at each copy number state, as determined by the SNP analysis. Mean and standard errors are shown.

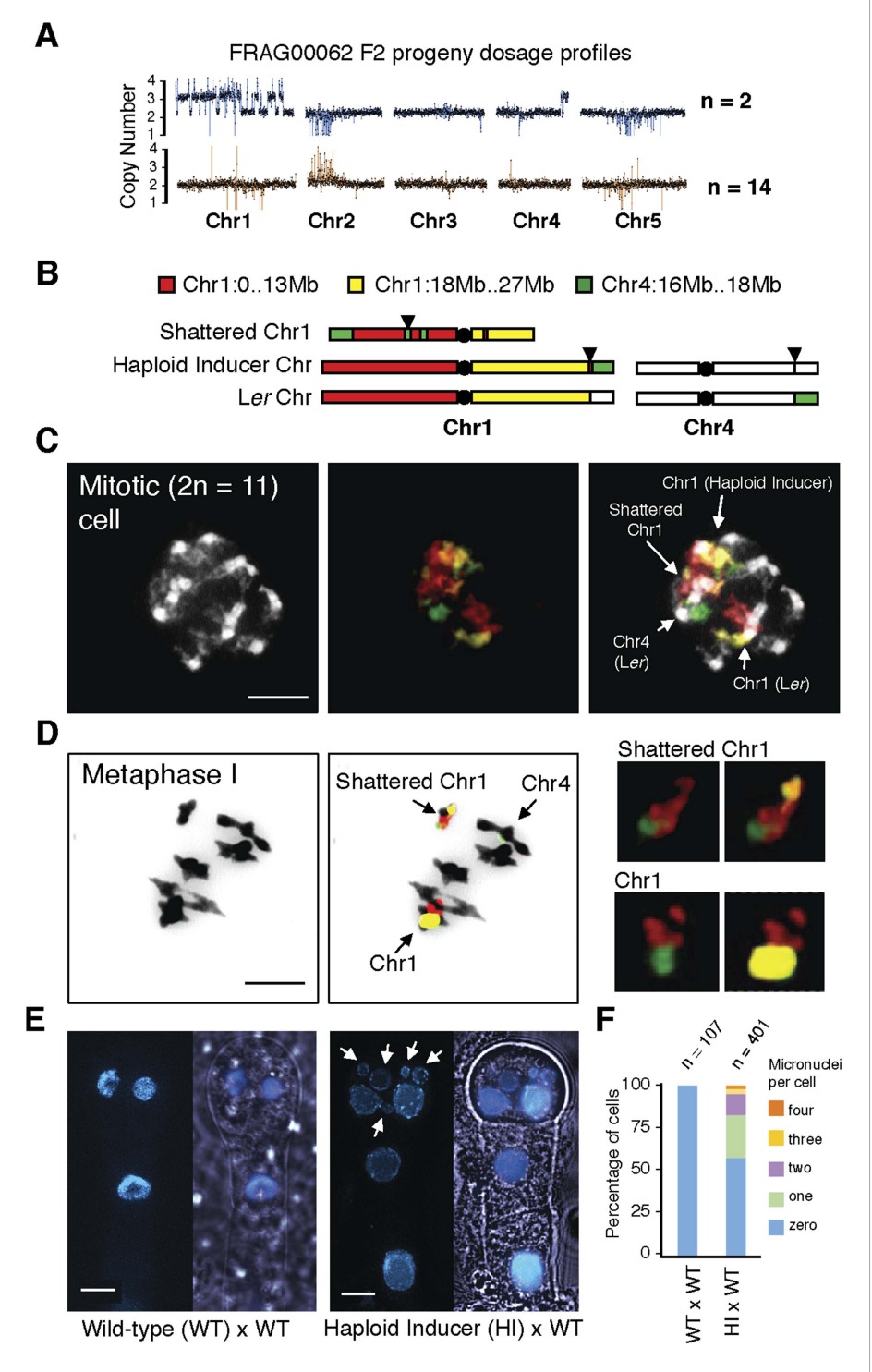

**Figure 4**. Stable inheritance and chromosome painting of a shattered aneuploid chromosome. (**A**) Dosage analysis from 16 F2 individuals from a selfed FRAG00062 individual. Progeny individuals either inherited the shattered chromosome intact (n = 2) or appeared diploid (n = 14). (**B**) Cartoon of the different versions of chromosomes 1 and 4 expected to be present in FRAG00062. Chromosome painting probes and corresponding chromosome positions used for (**C**) and (**D**) are shown. Black triangles indicate the position of the reciprocal Chr1/Chr4 translocation present in the haploid inducer line, whereas black circles indicate centromere positions. (**C**) A mitotic cell from FRAG00062 with 11 chromosomes, including four painted chromosomes. Scale bar = 5 µm. (**D**) The shattered Chr1 from FRAG00062 remains unpaired at meiosis as shown here at Metaphase I. Enlargements of the shattered Chr1 and

*Figure 4. Continued*

paired Chr1 are shown on the right. Scale bar = 5 μm. (**E**) Nuclei from a two-cell stage embryo from a wild-type cross (left panel) and from an embryo undergoing uniparental genome elimination (right panel). Nuclei are visualized using CFP-tagged histone H2B from the pollen parent superimposed with an image of the embryo visualized under light microscopy. Note the presence of micronuclei from the embryo undergoing genome elimination (right panel). Scale bar = 5 μm. (**F**) Percentage of micronuclei observed in wild-type crosses and genome elimination crosses. The different percentages of micronuclei per cell are indicated.

The following figure supplements are available for figure 4:

**Figure supplement 1**. Analysis of duplicated and triplicated blocks from FRAG00062.

**Figure supplement 2**. DNA FISH on the shattered aneuploid chromosome from FRAG00062.

---

(*Subrahmanyam and Kasha, 1973*; *Gernand et al., 2005*). We dissected embryos from a genome elimination cross and observed one to four micronuclei per cell (*Figure 4E, F*) in 81% of the embryos (n = 110), but none in embryos from control crosses (n = 21, p < 0.001). The presence of micronuclei suggests that sub-compartmentalized lagging chromosomes can be shattered by double stranded DNA breaks, reassembled haphazardly by non-homologous end joining (NHEJ), and finally restituted into the main nucleus (*Crasta et al., 2012*).

In order to reconstruct breakpoint junctions, we sequenced FRAG00062 to 100× coverage, extracted read pairs from the ends of duplicated and triplicated blocks and performed de novo assembly. 38 such junctions were assembled (*Supplementary file 4*) and a random subset of 12/12 were confirmed by PCR (*Figure 5—figure supplement 1*) followed by Sanger sequencing to demonstrate the accuracy of the de novo assembly. All reconstructed junctions were consistent with NHEJ with either microhomology, observed as 2–15 bp of sequence overlap (*Hastings et al., 2009*), blunt fusions, or unidentified sequence insertions (*Figure 5B*). We also observed inversions (fragments that join in head to head or tail to tail orientation) in 47% of our breakpoint junctions (*Supplementary file 4*). The size distributions of microhomology tracts and insertions are indicated in *Figure 5—figure supplement 2*.

Overall, triplicated block sizes from FRAG00062 were significantly smaller than duplicated blocks (n = 23 in both cases, with p < 0.001, *Figure 5C*) and these triplications cannot be easily explained from a missegregated chromosome. Duplicated and triplicated blocks could therefore, have different origins. To address this question, we asked whether breakpoint junctions of the two different copy number states display differential association to various genomic and chromatin features such as genes and repeated elements (*Lamesch et al., 2012*), DNA replication origins (*Costas et al., 2011*), DNase I hypersensitive sites (DHS) (*Zhang et al., 2012*) and nine non-overlapping chromatin states that partition the *Arabidopsis* genome (*Sequeira-Mendes et al., 2014*) (*Supplementary file 5*). When analyzing windows of 1000 bp centered around the breakpoints of duplicated blocks, we observed an enrichment in genic DNA (from 53% background level to 70%, p < 0.01, *Figure 5D,F*). A subtler, but still significant, increase was observed when using larger windows (10,000 bp , from 53% background level to 62%, p < 0.01, *Figure 5F*). Consistently, 42% of breakpoint junctions from FRAG00062 are predicted to generate chimeric gene products (*Supplementary file 4*). In the same analysis, we noted that the breakpoint regions of duplicated and triplicated blocks contained some genomic features that differed in frequency. In particular, replication origins, which occupy less than 1% of 10,000 bp windows around the borders of duplicated blocks, are present in almost 8% around the borders of triplicated blocks (compared to a genome average of 3.5%, p < 0.05, *Figure 5E,G*). The association of the breakpoints flanking duplicated DNA to genic DNA and of those flanking triplicated DNA to replication origins suggests the contribution of two distinct mechanisms to restructuring of the same chromosome (*Figure 6*). The first, chromothripsis acting through breakage and ligation (*Stephens et al., 2011*; *Korbel and Campbell, 2013*). The second, chromoanasynthesis, via replication fork collapse and template switching (*Hastings et al., 2009*; *Liu et al., 2011*; *Kloosterman and Cuppen, 2013*).

Our in silico reconstruction suggests that NHEJ is involved in repairing breaks that occurred on the shattered chromosomes. To test this explanation, we created a haploid inducer carrying

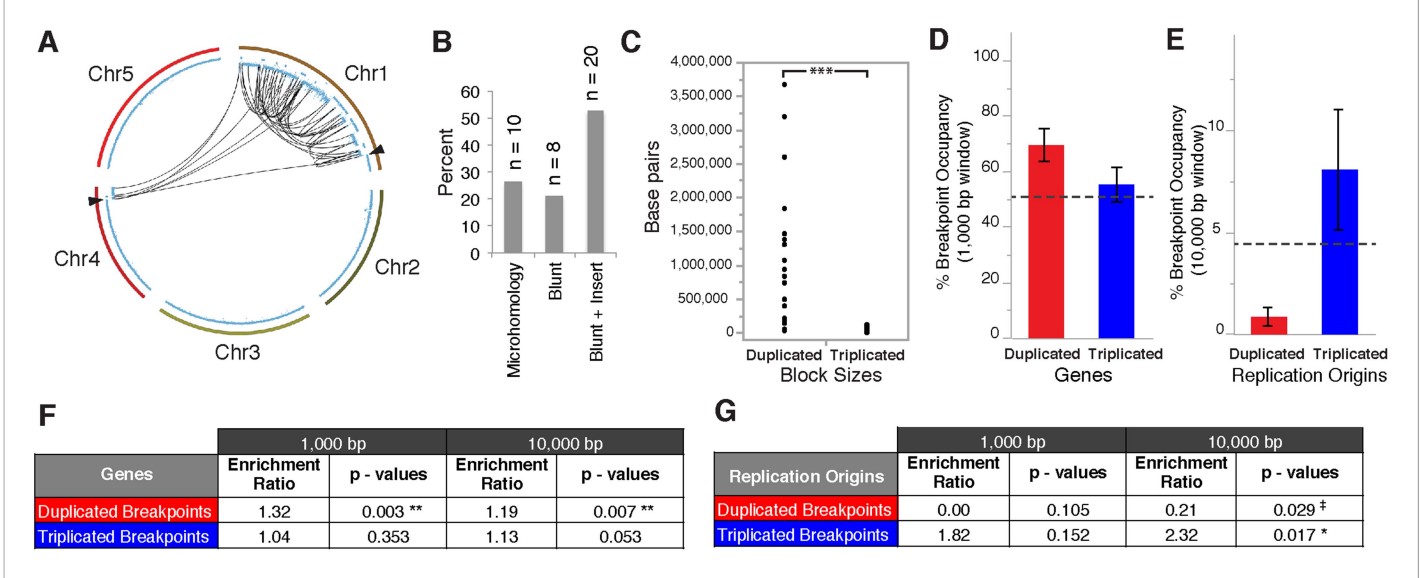

**Figure 5**. Breakpoint junctions and enriched features surrounding breakpoints of duplicated and triplicated blocks. (**A**) Plot of all chromosomes and corresponding dosage plots (blue line) from FRAG00062. Black curves depict novel junctions identified from genomic reconstruction of the shattered haploid inducer Chr1. Black triangles indicate the reciprocal translocation between Chr1 and Chr4 from the haploid inducer genome. (**B**) Percentage of junctions with 2–15 bp of microhomology, blunt junctions or junctions with unidentified sequence insertion observed from FRAG00062. (**C**) Plot of duplicated (n = 23) and triplicated (n = 23) block sizes from FRAG00062. (*** Student's $t$-test p < 0.001). (**D**, **E**) Occupancy of genes and replication origins around breakpoints regions, calculated using windows of 1000 bp or 10,000 bp centered on each breakpoint. Dashed horizontal lines indicate the genome-wide occupancy of each feature. Error bars indicate standard error. (**F**, **G**) Enrichment ratio of genes and replication origins (See methods for description of enrichment ratio). Genes (**F**) are significantly enriched surrounding duplicated breakpoints regardless of window size (1000 or 10,000 bp, **p < 0.01). For windows of 10,000 bp, replication origins (**G**) are significantly enriched at triplicated breakpoints (*p < 0.05) while significantly under-represented (‡p < 0.05) at duplicated breakpoints.

The following figure supplements are available for figure 5:

**Figure supplement 1**. PCR confirmation of breakpoint junctions for FRAG00062.

**Figure supplement 2**. Breakpoint junctions types from FRAG00062.

a homozygous null mutation in *LIG4* (DNA Ligase IV), a conserved component of the canonical NHEJ pathway. Pollinating it with wild-type *LIG4/LIG4* pollen (from L*er gl1-1*) resulted in normal haploid induction frequencies. However, when mutant *lig4-2/lig4-2* pollen was used, the frequency of haploids doubled at the expense of aneuploids and diploids (*Table 1* and *Figure 7*). This effect was still observed when the seed parent carried the WT allele (*Table 1*). It is possible that parental-specific haploinsufficiency results from early loss of the wild-type *LIG4* allele located on the chromosome targeted for elimination, which in this case is the maternal chromosome. This result indicates that NHEJ contributes to formation or persistence of aneuploid and diploid progeny and that unrepaired double-stranded DNA breaks increase elimination of the haploid inducer genome, similar to observations in mouse-human hybrid genome elimination (*Wang et al., 2014*). We hypothesize that missegregated chromosomes enter a degradative pathway initiated by endonucleolytic breaks. Occasionally, such chromosomes are rescued (i.e., restituted to a haploid or diploid nucleus) through a pathway requiring NHEJ, resulting in aneuploidy. Therefore, more haploids are produced when the NHEJ pathway is impaired (*Figure 6*).

## Discussion

Taken together, our results provide evidence for the occurrence of chromosome restructuring (*Cai et al., 2014*; *Morrison et al., 2014*) when diverged individuals hybridize, identifying a centromere-based mechanism for genomic instability. This phenomenon studied here depends on chimeric

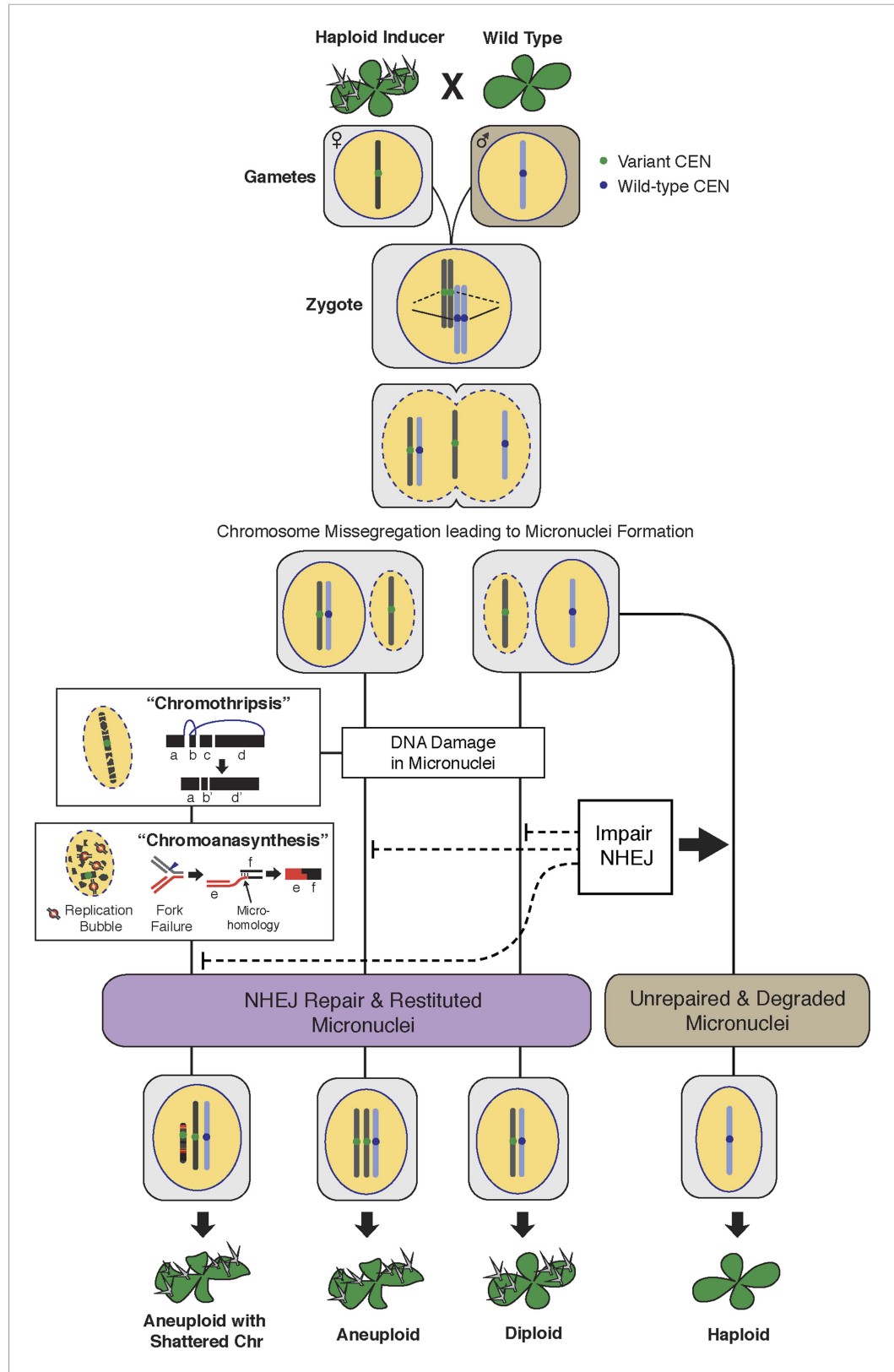

**Figure 6**. The process of genome elimination and connected models for chromosomal rearrangements. Genome elimination ensues when a haploid inducer expressing a variant CENH3 protein mates with the wild type. In many cases, the chromosomes marked by the variant CENH3 missegregate in the embryo and are compartmentalized in
*Figure 6. continued on next page*

*Figure 6. Continued*

micronuclei. DNA damage, NHEJ repair and restitution of the micronucleus to the euploid pole nucleus can result in aneuploidy or diploidy. Alternatively, shattered chromosomes result from chromothripsis and chromoanasynthesis. The former involves fragmentation and random ligation, the latter replication fork collapse and microhomology-mediated strand switching. As a consequence, the pulverized and reassembled chromosome forms a single unit and can be meiotically inherited. The schematics for chromothripsis and chromoanasynthesis are shown sequentially for convenience, but their order has not been determined. In addition, our results obtained using *DNA ligase4-2* mutants suggest that the NHEJ pathway plays an important role in the repair of the haploid inducer chromosomes that contribute to diploid and aneuploid progeny, such that when NHEJ is inhibited, haploid induction frequency increased.

---

CENH3, but a similar effect was observed when the haploid inducer strain expresses CENH3 of a close species (*Maheshwari et al., 2015*), indicating the effectiveness of natural and artificial variation. While the genesis and fate of restructured chromosomes is difficult to study in humans, their formation, effects, and even transmission in *Arabidopsis* are within experimental reach, as demonstrated by the enhancing effect of NHEJ mutants on haploid induction. The range of phenotypes, the formation of copy variants and of chimeric genes at junctions, and their occasional meiotic transmission, suggest that catastrophic chromosomal restructuring, could contribute to heritable genetic variation.

## Materials and methods

### Plant material and growth conditions

All plants were grown in Sunshine Professional Mix Peat-Lite Mix 4 (SunGro Horticulture, Agawam, MA) under 16hr/8hr light/dark photoperiod in a growth room set at 21˚C. F1 seeds from *GFP*-tailswap crosses were germinated on MS agar plates and 2-week old seedlings were transplanted into soil. The *lig4-2* (SAIL_597_D10) line used is in the Col-0 background. Genotyping primers (5′ to 3′) used to are lig4-2/LP2: GATATGACAAGCCTTGGCATGAATGT, lig4-2/RP: AAAGTGGATGACATCTCGCTG and LB1: GCCTTTTCAGAAATGGATAAATAGCCTTGCTTCC for the left border of the SAIL T-DNA insertion.

### Genomic DNA preparation, sequencing and read processing

All DNA samples were extracted from leaves using Nucleon Phytopure kits (GE Healthcare, Pittsburgh, PA). 1.5 µg of DNA were used for a PCR-free library preparation using the NEBNext DNA Library reagents with Nextflex-96 indexes (Bioo Scientific, Austin, TX) using a PCR-free protocol. 2 µl of each 96-barcoded libraries were pooled and sequenced using the 50 bp protocol on a single lane of Hiseq 2000 at the Vincent J. Coates Genomics Sequencing Laboratory at UC Berkeley. Demultiplexing was performed by the same facility and resulting raw reads were processed with a custom Python script (Filter_N_Adapter_Trim_Batchmode.py – available from GitHub repository: https://github.com/KorfLab/FRAG_project) that removes the filtered reads from Cassava 1.8, adapter sequences, reads that contain Ns and trims reads for quality.

**Table 1**. Haploid induction frequency from genome elimination crosses using *lig4-2* mutants

| Haploid Inducer ♀ | ♂ | Total | Aneuploid | Diploid | Haploid |
|---|---|---|---|---|---|
| *GFP*-tailswap | Ler *gl1* | 606 | 33% | 28% | 39% |
| *GFP*-tailswap | *lig4-2* | 148 | 8% | 9% | 83% * |
| *lig4-2 GFP*-tailswap | Ler *gl1* | 173 | 29% | 31% | 40% |
| *lig4-2 GFP*-tailswap | *lig4-2* | 159 | 14% | 5% | 81% * |

Haploid inducer lines or haploid inducer line with the *lig4-2* mutation were crossed to wild-type Ler *gl1* or *lig4-2* mutant pollen (*Student's *t*-test, p < 0.001).

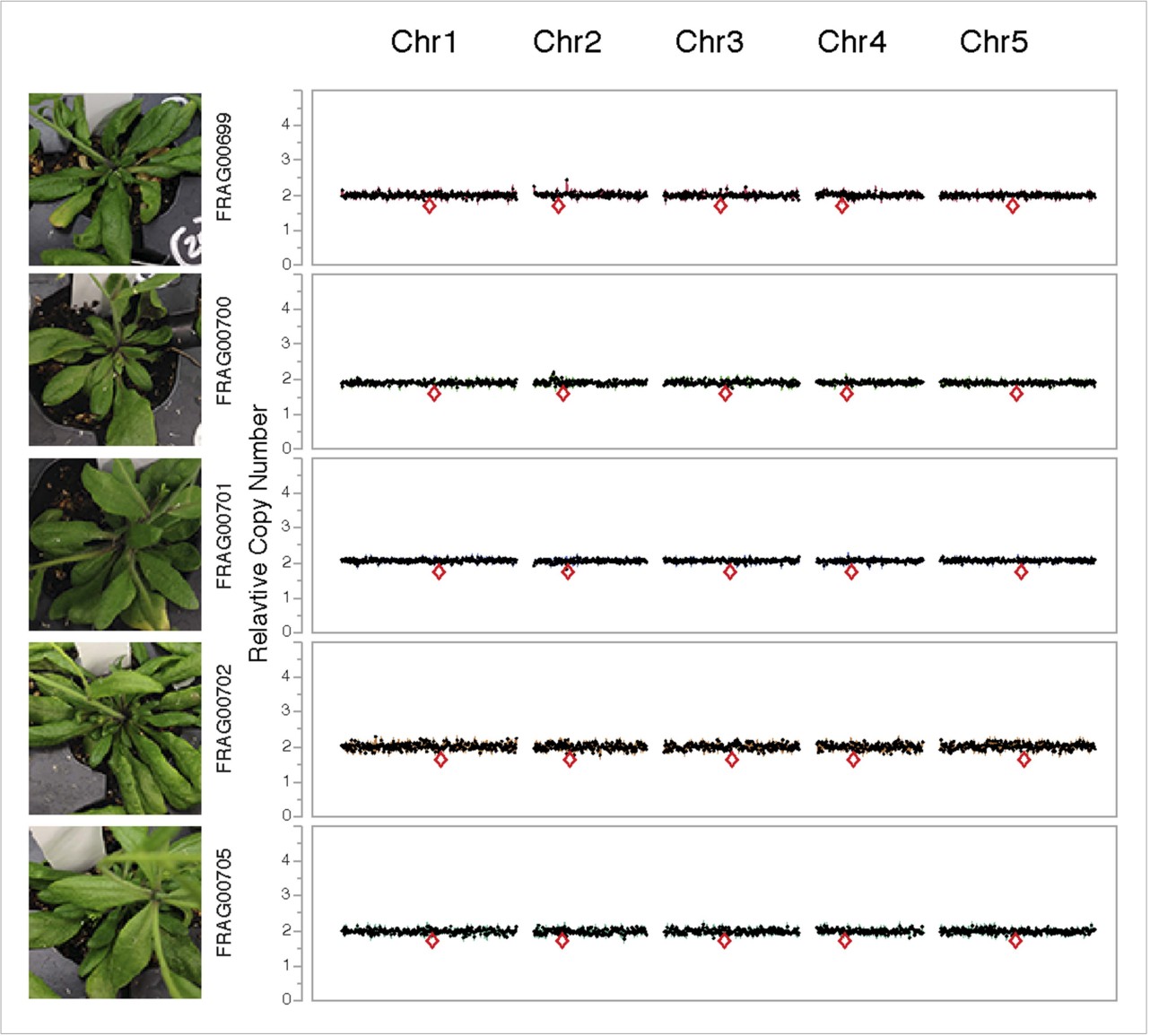

**Figure 7**. Dosage plots for *lig4-2* haploids isolated from a haploid induction cross using diploid *lig4-2* as the male donor. Dosage plots of *lig4-2* haploids based on 150 kbp non-overlapping bins across all five *Arabidopsis* chromosomes. Euploid chromosome dosage plots for *lig4-2* haploids have the appearance of having a copy number of 2 only because euploid chromosome dosage was calculated with the value of 2 in this analysis. Centromere positions are indicated by red diamonds.

## Chromosome dosage analysis

For dosage plot analyses, 50 bp single reads were mapped to the TAIR10 *A. thaliana* reference genome sequence using BWA (*Li and Durbin, 2009*) and default parameters. Dosage variation was detected as previously described (*Henry et al., 2010*), and is described in detail at Bio-protocol (*Tan et al., 2016*). The genomic reference chromosomes were partitioned into consecutive non-overlapping bins of 100,000 bp and the percentage of reads mapping to each bin from each sample was recorded. Relative coverage was calculated by dividing the percentage obtained for each bin by either the corresponding mean percentage for all individuals or the corresponding percentage for the control individual. The relative coverage was set at 2 to represent the diploid background copy value.

## SNP analysis

Positions polymorphic between Col-0 and L*er* were identified using sequencing reads from a diploid Col/L*er* hybrid control, a L*er* plant and a Col-0 plant using custom python scripts.

Specifically, polymorphic positions were first identified if they were covered at least 25 times in the hybrid reads and contained two alleles, each representing at least 40% of the allelic calls. Reads from the Col-0 and Ler parents were then used to assign alleles to the two parents. Positions were only retained if they were homozygous in both parents (represented at least 97% of the allelic calls) and covered at least 6 times in the Col-0 library and at least once in the Ler library. This process resulted in the identification of 107,640 SNP positions (*Supplementary file 6*). Next, reads from each of the samples were mined for allele calls at these positions and each read was assigned to one or the other parent based on the parental information. If the read did not match either allele, the genotype was reported as 'na'. Finally, genotype information was pooled by consecutive, non-overlapping bins of 1 Mb to derive a percentage of Ler allele per bin for each sample. Using this measure, the Col-0/Ler diploid hybrid is expected to exhibit 50% Col-0 across the genome.

## Cytogenetic analysis

All analyses were carried out using chromosome spreads from young anthers. BAC contigs specific for *A. thaliana* chromosomes 1 and 4 were used as painting probes. BAC DNA was labeled with biotin-, digoxigenin- or Cy3- deoxyuridine triphosphate by nick translation as previously described (*Lysak and Mandáková, 2013*). Labeled DNA probes were pooled, hybridized to suitable chromosome spreads and visualized using fluorescent microscopy. See *Supplementary file 7* for the list of BAC clones used as painting probes.

## Breakpoint assembly

Breakpoints from FRAG00062 were identified using a high-density 500 bp bin-size dosage plot produced using 50 bp reads extracted from 100 bp paired-end sequencing reads of the FRAG00062 library obtained from an Illumina HiSeq 2000 instrument. Blocks of duplicated or triplicated dosage were defined by eye. A custom script (batch-specific-junction-search.py – available from GitHub repository: https://github.com/KorfLab/FRAG_project) was used to extract the sequencing reads mapping within a 2000 bp region around each breakpoint. These sequences were then assembled using the PRICE genome assembler using the standard paired-end assembly setting (*Ruby et al., 2013*). Resulting contigs were aligned to the *Arabidopsis* reference genome by NCBI-BLASTN and characteristic breakpoint junctions were identified when two halves of a contig mapped discordantly to the reference genome. Primers flanking 12 randomly selected breakpoint junctions were designed using Primer3 (*Li and Durbin, 2009*) based on their respective de novo assembled contigs. Standard PCR procedures were used for amplification using oligo pairs (*Supplementary file 8*) and GoTaq Green Mastermix (Promega Corporation, Madison, WI) on 1 ng DNA from FRAG00062 and FRAG00080 (a diploid sibling control) followed by Sanger sequencing.

## Breakpoint analysis

The *A. thaliana* TAIR10 genome annotation includes genomic locations for various features in Generic Feature Format Version 3 (GFF). Files specifying genes, transposon, satellite repeats, and replication origins were downloaded from the TAIR FTP site (ftp://ftp.arabidopsis.org//Maps/gbrowse_data/TAIR10/). The GFF file containing the location of mapped replication origins was available from a study by *Costas et al., (2011)*. These GFF files were combined with results about mapped DHS (*Zhang et al., 2012*) and details from the recent work by *Sequeira-Mendes et al., (2014)*, which combined various published epigenomic studies to partition the entire genome into nine different chromatin states. Perl scripts were used to convert the DHS and chromatin state information into GFF format, and these scripts, along with the resulting combined GFF file are available from a GitHub repository: https://github.com/KorfLab/FRAG_project.

The set of genome features in the combined GFF file were compared to the annotated set of duplicated and triplicated blocks. Various Perl scripts available from the above GitHub repository, along with a GFF representation of all blocks were used to assess the enrichment of genomic features at the breakpoint regions of duplicated/triplicated blocks. Specifically, window sizes of either 1000 or 10,000 bp were centered on each breakpoint coordinate, and the number of bp contributed by each feature of interest were summed across all windows. We also calculated the number of bp contributed by each feature outside those windows.

Enrichment ratios were then calculated using the percentage of bases occupied by each feature across all windows at breakpoints compared to percentage of the same features that occupy the remaining fraction of the aneuploid chromosome. The p-values were determined by shuffling experiments in which the locations of the breakpoints were randomized 1000 times, with the resulting shuffled ratios compared to the ratios observed in the real data (*Supplementary file 5*).

## Acknowledgements

We thank Howard Hughes Medical Institute (HHMI) and Neil Hunter for their one-year support for EHT after the passing of SWLC We would also like to thank Anne B. Britt for the *lig4-2* line, Meric C Lieberman for customized scripts and Kathie J Ngo for assistance with data management. This work used the Vincent J. Coates Genomics Sequencing Laboratory at UC Berkeley, supported by NIH S10 Instrumentation Grants S10RR029668 and S10RR027303. This work also used the Caliper Sciclone NGS which was purchased with support from the NIH Shared Instrumentation Grant 1S10OD010786-01 awarded to LC. This work was funded by the HHMI and the Gordon and Betty Moore Foundation (GBMF) through grant GBMF3068 (to LC). Work by TM and MAL are funded by a research grant from the Czech Science Foundation (P501/12/G090) and by the European Social Fund (CZ.1.07/2.3.00/30.0037). MR is supported by DBT-Ramalingaswami fellowship. SWLC was a HHMI-GBMF Investigator who pioneered the field of centromere-mediated genome elimination until his untimely passing in 2012.

## Additional information

### Funding

| Funder | Grant reference | Author |
|---|---|---|
| Gordon and Betty Moore Foundation | GBMF3068 | Isabelle M Henry, Maruthachalam Ravi, Keith R Bradnam, Mohan PA Marimuthu, Ian Korf, Luca Comai |
| Howard Hughes Medical Institute (HHMI) | GBMF3068 | Isabelle M Henry, Maruthachalam Ravi, Keith R Bradnam, Mohan PA Marimuthu, Ian Korf, Luca Comai |
| Czech Science Foundation | P501/12/G090 | Terezie Mandakova, Martin A Lysak |
| European Social Fund | CZ.1.07/2.3.00/30.0037 | Terezie Mandakova, Martin A Lysak |
| Department of Biotechnology, Ministry of Science and Technology | Ramalingaswami fellowship | Maruthachalam Ravi |

The funders had no role in study design, data collection and interpretation, or the decision to submit the work for publication.

### Author contributions

EHT, Conception and design, Acquisition of data, Analysis and interpretation of data, Drafting or revising the article; IMH, LC, Conception and design, Analysis and interpretation of data, Drafting or revising the article; MR, Conception and design, Acquisition of data, Analysis and interpretation of data; KRB, IK, Analysis and interpretation of data, Drafting or revising the article; TM, MPAM, Acquisition of data, Analysis and interpretation of data; MAL, Acquisition of data, Analysis and interpretation of data, Drafting or revising the article; SWLC, Conception and design, Analysis and interpretation of data

### Author ORCIDs

Ek Han Tan, http://orcid.org/0000-0001-8872-7404
Isabelle M Henry, http://orcid.org/0000-0002-6796-1119
Keith R Bradnam, http://orcid.org/0000-0002-3881-294X
Luca Comai, http://orcid.org/0000-0003-2642-6619

# Additional files

## Supplementary files

• Supplementary file 1. List of the aneuploid individuals from *GFP*-tailswap × L*er gl1-1* crosses used in this study. Unique identifiers (FRAG IDs) are listed for each individual, along with the *GFP*-tailswap parental lines as well as their corresponding aneuploid chromosome types. The final 16 columns indicate whether individuals exhibited the numerical, truncated or shattered aneuploid chromosome type, and which specific chromosomes displayed these features.

• Supplementary file 2. List of aneuploid individuals obtained from selfed Col-0 triploids. Unique identifiers (FRAG IDs) are listed for each individual and indicate which specific chromosomes displayed the numerical aneuploid chromosome type.

• Supplementary file 3. List of diploid and aneuploid individuals obtained from selfed *GFP*-tailswap. Unique identifiers (FRAG IDs) are listed for each individual and indicate which specific chromosomes displayed the numerical aneuploid chromosome type.

• Supplementary file 4. Breakpoint junctions from FRAG00062. The positions and types of repair for each breakpoint junction are indicated, along with their orientation as well as possible outcomes for a novel gene product.

• Supplementary file 5. Enrichment ratios of genomic features surrounding breakpoints from duplicated or triplicated blocks. The ratios of genomic features listed in column 1 are calculated using 1000 bp and 10,000 bp windows surrounding breakpoints from duplicated or triplicated blocks. Regions with enriched ratios has a value of > 1 while regions that are under-represented have values of < 1. Light orange boxes are indicated for p < 0.05 and light red boxes for p < 0.01.

• Supplementary file 6. Single nucleotide polymorphisms between Col-0 and L*er*. A list of SNPs from Col-0 and L*er* used in this study and their chromosomal positions.

• Supplementary file 7. BAC clones used for chromosome painting. Listing of the corresponding chromosome, BAC clone and associated GenBank accession used.

• Supplementary file 8. Oligonucleotides sequences for analysis of breakpoint junctions from FRAG00062.

## Major dataset

The following dataset was generated:

| Author(s) | Year | Dataset title | Dataset ID and/or URL | Database, license, and accessibility information |
|-----------|------|---------------|-----------------------|-------------------------------------------------|
| Tan EH, Comai L | 2014 | SRP051357 | http://www.ncbi.nlm.nih.gov/bioproject/PRJNA269994 | Publicly available at NCBI Sequence Read Archive (PRJNA269994). |

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
