## [Decision Letter]

Thank you for sending your work entitled “Catastrophic chromosomal restructuring during genome elimination in plants” for consideration at *eLife*. Your article has been favorably evaluated by Detlef Weigel (Senior editor) and three reviewers, one of whom is a member of our Board of Reviewing Editors.

The following individuals responsible for the peer review of your submission have agreed to reveal their identity: Bernard de Massy (Reviewing editor and peer reviewer) and Jim Haber (peer reviewer). A further reviewer remains anonymous.

The Reviewing editor and the other reviewers discussed their comments before we reached this decision, and the Reviewing editor has assembled the following comments to help you prepare a revised submission.

This manuscript presents an analysis of chromosome reorganization and elimination in crosses of *Arabidopsis* plants with different CENH3 variants. Although this particular study principally involves *GFP*-tailswap haploid-inducer lines, it has clear relevance to general biological phenomena resulting from hybridization of related species and aneuploidy. The authors have used cytogenetics complemented by a NGS approach to carry out a clear and coherent analysis of chromosome elimination and reorganization at the genome-wide level in these plants. This is a most interesting paper that follows up on a recent PLoS Genetics paper that showed that crossing plants with differing CENH3 N-terminal tails frequently resulted in genome instability. The present paper extends these studies and definitively analyzes the nature of the chromosome shattering, showing that the instability most likely arises in post-meiotic mitoses and likely most involves two different processes are implicated.

The data appear to be solid, are presented clearly and support the authors' conclusions. Overall the text is concise, clearly written and referenced. However several aspects need to be clarified. In particular, analyses of the genomic features at breakpoints and the role of *lig4*.

Major comments:

1) Presentation of the events: It is quite difficult to get a clear picture of the relationship between the different classes of aneuploidy and the genome rearrangements. An improved presentation of the data is necessary: A more precise description of the numerical class should be provided in particular with presentation of frequency of trisomies and monosomy and the phenotypes. The category of diploids (25%) appears to be based only the wild type phenotype? A diploid sample should be analyzed for copy number variation and rearrangements as control.

2) The experiment performed to address the meiotic origin is not convincing: The key experiment is to test progeny of *GFP*-tailswap. [27] indicate that this line is not fully fertile suggesting a meiotic defect. In addition, aneuploids are detected in the progeny of selfed plants. With the current information, the authors cannot exclude a meiotic origin of the truncated and shattered chromosomes identified in aneuploids.

3) Genomic features at breakpoints: The analysis of genomic features is very weak and not convincing (Figure 4). This analysis should be removed or validated by statistically tests. The abstract should be modified accordingly.

4) The effect of *Lig4* should be clarified: why such an effect when heterozygous in L*er* ecotype? What is the genotype/genomic structure of the haploids? If the absence of NHEJ leads to loss of recovery of shattered chromosomes, why is the proportion of haploids (relative to diploids) increased? How does it fit with the model presented in Figure 5 is unclear.

---

## [Author Response]

*1) Presentation of the events: It is quite difficult to get a clear picture of the relationship between the different classes of aneuploidy and the genome rearrangements. An improved presentation of the data is necessary: A more precise description of the numerical class should be provided in particular with presentation of frequency of trisomies and monosomy and the phenotypes*.

We have improved the presentation of the different classes of aneuploidy by including an example of each aneuploid type in a main figure (Figure 2), accompanied by a description in the text. The frequencies of the various primary trisomies and monosomies are now included in the text as well.

*The category of diploids (25%) appears to be based only the wild type phenotype? A diploid sample should be analyzed for copy number variation and rearrangements as control*.

We have now included dosage analysis of 10 individuals from a haploid induction cross, but identified as diploid visually. Consistent with their phenotype, they do not display any copy number variation or rearrangement (Figure 1—figure supplement 1).

*2) The experiment performed to address the meiotic origin is not convincing: The key experiment is to test progeny of* GFP*-tailswap.*
[27]
*indicate that this line is not fully fertile suggesting a meiotic defect. In addition, aneuploids are detected in the progeny of selfed plants. With the current information, the authors cannot exclude a meiotic origin of the truncated and shattered chromosomes identified in aneuploids*.

We agree that this is a possibility. Therefore, we sequenced 96 progeny individuals from a selfed *GFP*-tailswap. Of 96 sequences individuals, 94 were diploid and 2 were numerical aneuploids (Figure 2—figure supplement 4). No truncated and shattered aneuploids were observed, consistent with our results from the selfed triploid population and the hypothesis that chromosomal breaks originate from mitotic defects post-fertilization.

*3) Genomic features at breakpoints: The analysis of genomic features is very weak and not convincing (*Figure 4*). This analysis should be removed or validated by statistically tests. The abstract should be modified accordingly*.

We have repeated the analysis and validated it by statistical testing. We provide the corresponding data in the revised manuscript (Figure 5 and [Supplementary-material SD5-data]).

*4) The effect of* Lig4 *should be clarified: why such an effect when heterozygous in L*er *ecotype*?

We expanded the discussion of the *lig4-2* mutation effect including a hypothesis on the potential mechanism: “This effect was still observed when the seed parent carried the wild-type allele (Table 1). It is possible that parental-specific haploinsufficiency results from early loss of the wild-type *LIG4* allele located on the chromosome targeted for elimination, which in this case is the maternal chromosome.”

*What is the genotype/genomic structure of the haploids*?

Because the null *lig4-2* mutant used in the cross is in the Col-0 background, we could not use SNP after visual identification of *lig4-2* haploids to determine their genome's parental origin. Instead, the genotype of the *lig4-2* haploids was assessed by PCR using primers (included now in Materials and methods) that distinguish the T-DNA tagged *lig4-2* knock-out allele from the wild-type *LIG4* allele. This confirmed the absence of the latter allele, consistent with genome elimination having occurred as expected. We next performed dosage analysis on these individuals and observed that all tested were euploid (Figure 7).

*If the absence of NHEJ leads to loss of recovery of shattered chromosomes, why is the proportion of haploids (relative to diploids) increased? How does it fit with the model presented in*
Figure 5
*is unclear*.

We have expanded the explanation of this observation: “We hypothesize that missegregated chromosomes enter a degradative pathway initiated by endonucleolytic breaks. Occasionally, such chromosomes are rescued (i.e. restituted to a haploid or diploid nucleus) through a pathway requiring NHEJ, resulting in aneuploidy. Therefore, more haploids are produced when the NHEJ pathway is impaired.”